# SpeedyZero: Mastering Atari with Limited Data and Time

**Yixuan Mei**[1,2*]**, Jiaxuan Gao**[1,2*]**, Weirui Ye**[1]**, Shaohuai Liu**[1]**, Yang Gao**[1,2†]**, Yi Wu**[1,2†]

[1] Institute for Interdisciplinary Information Sciences, Tsinghua University, [2] Shanghai Qi Zhi Institute
`meiyixuan2000@gmail.com`

## Abstract

Many recent breakthroughs of deep reinforcement learning (RL) are mainly built upon large-scale distributed training of model-free methods using millions to billions of samples. On the other hand, state-of-the-art model-based RL methods can achieve human-level sample efficiency but often take a much longer overall training time than model-free methods. However, high sample efficiency and fast training time are both important to many real-world applications. We develop SpeedyZero, a distributed RL system built upon a state-of-the-art model-based RL method, EfficientZero, with a dedicated system design for fast distributed computation. We also develop two novel algorithmic techniques, Priority Refresh and Clipped LARS, to stabilize training with massively parallelization and large batch size. SpeedyZero maintains on-par sample efficiency compared with EfficientZero while achieving a $14.5\times$ speedup in wall-clock time, leading to human-level performances on the Atari benchmark within 35 minutes using only 300k samples. In addition, we also present an in-depth analysis on the fundamental challenges in further scaling our system to bring insights to the community.

## 1 Introduction

Deep reinforcement learning (RL) has achieved significant successes in the past few years. Prior work has scaled model-free RL training to computing clusters with tens to hundreds of machines, achieving human-level performance or beating human experts on various complex problems (Jaderberg et al., 2019; Baker et al., 2019; Berner et al., 2019; Vinyals et al., 2019). There are two fundamental ideas behind their successes: (1) training with larger batches for faster convergence, as used in the task of hide-and-seek (Baker et al., 2019), DOTA 2 (Berner et al., 2019) and even in many popular PPO projects (Yu et al., 2021; Stooke & Abbeel, 2018), (2) developing systems with high scalability, such as Gorila (Nair et al., 2015), Ape-X (Horgan et al., 2018), IMPALA (Espeholt et al., 2018) and R2D2 (Kapturowski et al., 2018), which can efficiently simulate huge numbers of environments in parallel. Despite the achievements, these model-free-RL applications consume an extremely high volume of samples, which can be infeasible for many real-world scenarios without an efficient simulator accessible.

By contrast, model-based RL methods require substantially fewer samples to train a strong agent. In particular, some recent works have even achieved comparable sample efficiency to humans in complex RL domains like Atari (Ye et al., 2021) or robotic control (Wu et al., 2022). The downside of model-based RL methods is that they often require a long training time (Schrittwieser et al., 2020; Ye et al., 2021). Although people have tried to accelerate simple model-based RL methods in the existing literature (Zhang et al., 2019; Abughalieh & Alawneh, 2019), state-of-the-art sample-efficient model-based RL such as EfficentZero (Ye et al., 2021), which requires complicated model learning and policy planning, are still time-consuming to run.

In this paper, we aim to build a state-of-the-art sample-efficient model-based RL system that trains fast in wall-clock time. To do so, we start with EfficientZero, a state-of-the-art sample efficient model-based RL method, and then accelerate it with massively parallel distributed training. We remark that scaling state-of-the-art model-based RL methods like EfficientZero is non-trivial for

---

*Equal Contribution
†Equal Advising

two major challenges. First, different from model-free RL, for which massive parallelization can be simply achieved by simultaneously simulating more environments, a single training step in EfficientZero requires substantially more computation steps, including model expansion, back-track search/planning, Q-value backup, and re-analyzing past samples. Therefore, it is non-trivial to efficiently parallelize these components. Second, we empirically notice that when the data producing rate is largely accelerated via parallelization and the batch size is increased, in order to retain the same sample efficiency, EfficientZero training may suffer from significant instabilities.

We present SpeedyZero, a distributed model-based RL training system, which leverages dedicated system-level optimization to largely reduce computation overhead while inheriting the high sample efficiency from EfficientZero. From the system perspective, SpeedyZero contains three major innovations for a much faster per-iteration training speed, including (1) a non-trivial partition over computation workload to reduce network communications, (2) applying shared memory queues for high-through-put and low-latency intra-process communication, and (3) an optimized data transfer scheduling with reduced CPU-GPU communication and redundant data transmission. Furthermore, from the algorithm perspective, SpeedyZero is equipped with two novel techniques, Priority Refresh (P-Refresh) and Clipped LARS, to significantly stabilize model-based training in the case of massive parallelization and larger batch sizes. P-Refresh is a distributed and more aggressive variant of prioritized experience replay (Schaul et al., 2015), which actively re-computes the *accurate* priorities of *all* the samples in the replay buffer. Clipped LARS is a variant of LARS (You et al., 2017) to ensure stable training with large batch size. The proposed techniques are shown to be critical for the overall success of SpeedyZero.

We evaluate SpeedyZero on the Atari 100k benchmark (Kaiser et al., 2019), SpeedyZero achieves human-level performance with only 35 minutes of training and 300k samples. Compared with EfficientZero, which requires 8.5 hours of training, SpeedyZero retains a comparable sample efficiency while achieving a $14.5\times$ speedup in wall-clock time. Ablation studies are also presented to show the effectiveness of each design component and technique in our system. In addition, we also conduct a further study on the effect of batch size, and the results show that when the batch size increases, SpeedyZero may significantly drop. We carefully analyze the underlying bottleneck and hope the insights can benefit the community for future research.

Our main contributions are summarized as follows,

- We develop SpeedyZero, a massively parallel distributed model-based RL training system featuring high sample efficiency and fast training speed. SpeedyZero masters Atari in 35 minutes with 300k samples, achieving $14.5\times$ speed up and on-par sample efficiency compared with the state-of-the-art EfficientZero algorithm.
- SpeedyZero adopts three system optimization techniques that significantly reduce training latency and improve training throughput and further leverages two algorithmic techniques, Priority Refresh and Clipped LARS, to stabilize training with massive parallelization and larger batch sizes.
- We present comprehensive ablation studies on our system components and in-depth analysis on the existing bottlenecks when further scaling SpeedyZero, leading to practical suggestions and open questions for the community.

## 2   RELATED WORK

**Distributed Machine Learning** With the emergence of larger datasets and larger models, distributed machine learning systems proliferate in industry and in research. Two main branches exist in this field: data parallelism and model parallelism. Data parallelism partitions an enormous dataset into small chunks computationally tractable on single machines (or GPUs) and assigns the chunks to different machines (or GPUs) in the training cluster. Successful frameworks of data parallelism include three generations of parameter servers (Smola & Narayanamurthy, 2010; Dean et al., 2012; Li et al., 2014) and distributed data parallel (Li et al., 2020). On the other hand, model parallelism like Megatron-lm (Shoeybi et al., 2019) handles the problem of training gigantic models with billions of parameters by assigning different layers of the model to different machines. In our case, the model is small enough to fit onto a single GPU, while the sample batch is too large for efficient single GPU training. Therefore, we use distributed data parallel provided by PyTorch (Li et al., 2020) for multi-GPU training in SpeedyZero.

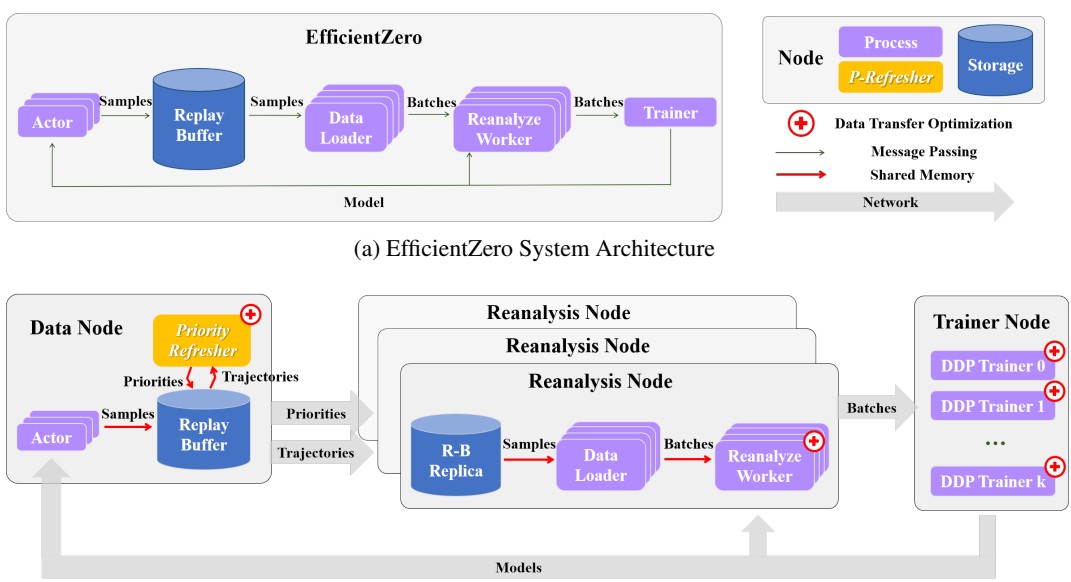

(a) EfficientZero System Architecture

(b) SpeedyZero System Architecture

Figure 1: **System Architecture Comparison between EfficientZero and SpeedyZero:** EfficientZero finishes all computation on a single machine. In comparison, SpeedyZero partitions the workflow into data collection (Data Node), batch reanalysis (Reanalysis Node), and training (Trainer node) and distributes the three stages to different machines. SpeedyZero has three main novelties in its architecture (Sec. 4.2): (1) modular design with non-trivial workload partition, (2) efficient on-node communication with shared memory, (3) data transfer optimizations that reduce CPU-GPU communication and overlap data transfer with computation. It also implements our proposed Priority Refresh (Sec. 4.3) in priority refresher of the data node for more stable training.

**Distributed Deep Reinforcement Learning** There have been many successful attempts to scale out model-based deep RL methods with distributed training. Assuming known environment models, AlphaGo (Silver et al., 2016) achieves super-human performance in the game of Go by distributing the rollout process in the Monte-Carlo Tree Search (MCTS) over hundreds of machines. Its successor MuZero (Schrittwieser et al., 2020) achieves astonishing results in the game of Go with thousands of TPUs. (Zhang et al., 2019) proposes an asynchronized framework for paralleling model-based RL training using the idea of parameter servers. However, their method is based on a simple model-based RL method, which is easy to parallelize but has relatively low sample efficiency. In the domain of model predictive control, many works (Abughalieh & Alawneh, 2019) study parallelism for improved speed, but focus on simple settings that are far less complicated than games like Atari. There are also many efforts in scaling out model-free deep RL methods, including Gorila (Nair et al., 2015), Ape-X (Horgan et al., 2018), IMPALA (Espeholt et al., 2018) and R2D2 (Kapturowski et al., 2018). These prior works typically focus on data-rich settings, requiring millions to billions of samples and hours to days of training. (Stooke & Abbeel, 2018) studies accelerated training of model-free methods such as PPO and A2C on a single machine. Our focus is on speeding up the training of model-based RL methods while maintaining high sample efficiency, which could bridge the training speed gap between model-based and model-free methods.

Besides system design optimizations, many prior works also adopt algorithmic improvements for better performance in distributed training of deep RL agents. IMPALA (Espeholt et al., 2018) introduces V-trace to correct the policy lag between the actors and learners. R2D2 (Kapturowski et al., 2018) proposes 'burn-in' steps to deal with the parameter lag in the recurrent neural networks. Many works also adopt prioritized experience replay (PER) in distributed training settings (Horgan et al., 2018; Kapturowski et al., 2018; Schrittwieser et al., 2020). Different from previous PER methods, our proposed prioritized sampling method, Priority Refresh, is able to stabilize value function training in the case of limited training data and training steps.

## 3 PRELIMINARY

**Reinforcement Learning.** A partially observable Markov decision process (POMDP) is defined by $M = \langle \mathcal{S}, \mathcal{A}, T, U, \Omega, \mathcal{O} \rangle$, where $\mathcal{S}$ is a set of states, $\mathcal{A}$ is the set of possible actions, $T$ is a transition

function over next states given the actions at current states, and $U : \mathcal{S} \times \mathcal{A} \to \mathbb{R}$ is the reward function. $\Omega$ is the set of observations of the agent and $\mathcal{O}$ maps states to probability distributions over observations. We use $o_{\leq t}$ to denote the history of observations at timestep $t$. The objective of the agent is to find a policy $\pi$ that maximizes the expected discounted return $\mathbb{E}[\sum_t \gamma^t u_t | a_t \sim \pi(\cdot | o_{\leq t})]$ where $u_t$ is the reward at step $t$.

**MuZero.** MuZero (Schrittwieser et al., 2020) is a model-based RL method based on the Monte-Carlo Tree Search (MCTS) algorithm. MuZero learns the environment dynamics and performs MCTS over the learned environment model to find a better policy. More specifically, MuZero models the environment with a representation function $h$, a dynamics function $g$, and a prediction function $f$. To find a high-quality policy given a history of observations $o_{\leq t}$, MuZero first encodes the observation history by $s_t^0 = h(o_{\leq t})$, which is used as the latent state at the root of the tree. To perform MCTS, MuZero runs $N$ simulation steps. In the $k$-th simulation step, a leaf node $s'$ and an unexplored action $a'$ on the leaf node are chosen by employing the UCT rule (Kocsis & Szepesvári, 2006; Rosin, 2011). Then a node expansion step comes by computing the next latent state $s_t^{k+1}$ and reward $r_t^{k+1}$ by the dynamics function $g$: $s_t^{k+1}, r_t^{k+1} = g(s', a')$, as well as the policy and value at $s_t^{k+1}$ by the prediction function $f$: $v_t^{k+1}, p_t^{k+1} = f(s_t^{k+1})$. At the end of each simulation step, the value $v_t^{k+1}$ is back-propagated along the tree path to update the Q values. The MCTS process is computationally expensive since it requires extensive CPU operations to do tree search as well as GPU resources for model inference.

MuZero interacts with the environment by searching a policy using MCTS over the learned environment model. The trajectories of data are then stored in the replay buffer. During training, a batch of observation histories are sampled and MuZero rolls out the environment model on the batch of observation histories $\{o_{\leq t}\}$ along the actions $\{a_{t...t+K-1}\}$ at the following $K$ steps, and predicts a batch of rewards $\{r_t^{1...K-1}\}$, policies $\{\pi_t^{0...K-1}\}$ and values $\{v_t^{0...K-1}\}$. To learn the models, the following loss is minimized,

$$\sum_k \mathcal{L}(u_{t+k}, r_t^k) + \lambda_1 \mathcal{L}(\pi_{t+k}, p_t^k) + \lambda_2 \mathcal{L}(z_{t+k}, v_t^k)$$

where $u_{t+k}$ is the environment reward, $\pi_{t+k}$ is the target policy obtained through MCTS over a target model, $z_{t+k} = \sum_{i=0}^{n-1} \gamma^i u_{t+k+i} + \gamma^n v'_{t+k+n}$ is the discounted $n$-step return, $v'_t$ is the value computed by the target model.

To improve the sample efficiency, MuZero Reanalyze algorithm (Schrittwieser et al., 2020) re-generates the policy and value of a training batch when the batch is sampled from the replay buffer. Compared with MuZero, MuZero Reanalyze uses significantly fewer samples while still achieving strong results. It is worth noting that the reanalysis step over the sampled batch is computationally expensive since it involves an additional MCTS procedure.

**EfficientZero.** EfficientZero (Ye et al., 2021) is a sample-efficient visual RL algorithm built on top of MuZero Reanalyze algorithm, which re-computes the target policies via MCTS when a training batch is sampled from the replay buffer. EfficientZero further proposes several augmentations in visual RL tasks, including using self-supervised consistency loss to provide more training signals to the environment model, predicting the value prefix instead of the reward to deal with aleatoric uncertainty and off-policy correction for the $n$-step return. The workflow of EfficientZero is shown in Fig. 1a, where the Reanalyze workers continuously generate the training batch. EfficientZero suffers from the same computation expense issue as MuZero Reanalyze due to the reliance on reanalyzing batches. Our method SpeedyZero inherits the sample efficiency optimizations from EfficientZero and boosts its training speed by $14.5\times$.

## 4 SpeedyZero: A Fast and Efficient Model-Based RL System

### 4.1 Overview

The ultimate goal of SpeedyZero is to speed up the training of EfficientZero-based RL agents while maintaining on-par sample efficiency. We achieve this through efforts on both the system side and algorithm side in SpeedyZero, as shown in Fig. 1. The system optimizations in SpeedyZero help us reduce the time needed for each training step. The algorithm optimizations reduce the number

of training steps needed while maintaining the stability of the training process. We will discuss the system optimizations in Sec. 4.2 and the algorithm optimizations in Sec. 4.3 respectively.

## 4.2 SPEEDYZERO SYSTEM DESIGN

As shown in Fig. 1, SpeedyZero partitions the workflow of EfficientZero into three stages: data collection, batch generation, and training. The three stages are distributed to the data node, the reanalysis node, and the trainer node respectively. SpeedyZero features the following three system design novelties for higher training step throughput and lower latency on critical modules.

**Modular Design and Non-Trivial Workload Partition:** A naive partition of workload to multiple machines entails massive network data transfer, inducing high latency and low throughput. SpeedyZero follows a modular design, in which we partition the workflow into three major stages so that the data transfer across stages is reduced and we assign one type of node for each stage. The machines (nodes) in SpeedyZero work together asynchronously and we can easily add more machines for a single type of node if it becomes the bottleneck. Our partition strategy helps us scale out SpeedyZero for higher throughput with low network latency overhead.

**Efficient On-Node Communication with Shared Memory:** Tradition message passing communication between different processes on the same machine serializes data on the sender side and deserializes them again on the receivers (known as data ser/des). This process also entails multiple memory copies. The data ser/des and memory copies make message passing extremely slow when the amount of data we need to transfer is huge. We notice that in our settings, the majority of data transferred can be expressed using NumPy arrays and many of these arrays are written once but read multiple times. Therefore, we develop a special shared memory object store for on-node communication. It avoids data ser/des for NumPy arrays and supports non-copy, lock-free reads for them. With this shared memory object store in hand, processes in SpeedyZero can communicate with higher bandwidth and lower latency.

**Data Transfer Optimizations:** We empirically find that batch generation latency on the reanalysis node and priority refresh latency on the data node affect the final performance a lot. We also notice that a considerable amount of time is spent on CPU-GPU data transfer in the two components. Therefore, we reduce the CPU-GPU data transfer during batch reanalysis by storing all MCTS latent states on GPU. For priority refresh (Sec. 4.3), we store the observations in the replay buffer on GPU to avoid loading them every time the priorities are re-computed. These two optimizations help SpeedyZero better utilize GPU VRAM to reduce the latency on critical components.

Data transfer can also overlap with computation in many cases to further reduce latency and improve throughput. In SpeedyZero, we overlap network transmission with computation by sending and receiving network packages in separate processes, allowing workers to continue their jobs when the package is flying through the network. Moreover, on the trainer node, we also overlap batch loading with training by preloading the next batch into GPU.

## 4.3 ALGORITHM IMPROVEMENTS

**Unstable Model-Based Training:** We empirically find that the predicted values may behave unstably, especially during the initial training stage. The instability issue is largely due to the accelerated training speed and a reduced amount of training steps due to the requirement of unchanged sample efficiency. As shown in Fig. 2, the predicted values climb abnormally high at the beginning, and it takes many steps before the values decrease back to the normal range. Notably, this phenomenon exists when using either prioritized experience replay (DPER) (Horgan et al., 2018) or uniform sampling from the replay buffer. The surge in predicted values at the beginning prevents proper policy improvements since MCTS relies on predicted values. Also, when scaling SpeedyZero to a larger batch size, we observe several sudden large gradients during training across a wide range of trials, as shown in Fig. 3a. We remark that the issues are not severe in EfficientZero since EfficientZero uses a much longer overall training time and a smaller batch size than SpeedyZero does.

**Priority Refresh:** To address the issue of unstable values, we propose *Priority Refresh*, in which we actively refresh the priorities of all data points in the replay buffer. As shown in Fig. 1b, a group of *priority refreshers* periodically update the priorities of all data points in the replay buffer. The latest priorities are synchronized to all reanalysis nodes with a constant frequency. Since the goal is to stabilize the values, we use TD errors as the priorities. The key difference between P-Refresh and

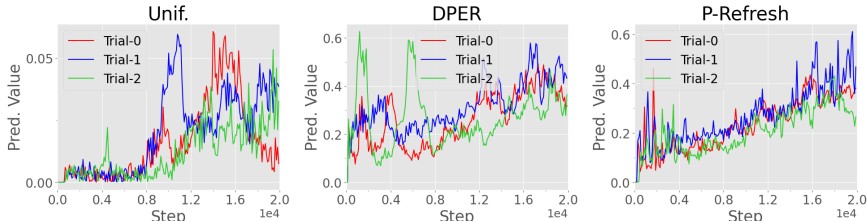

Figure 2: The predicted values of different trials when using uniform sampling from the replay buffer (Unif.), distributed prioritized experience replay (DPER), and priority refresh (P-Refresh) in Jamesbond. Uniform sampling exhibits very unstable values. Values of DPER are less unstable but still suffer from the same instability issue. In contrast, P-Refresh shows stable improvement in the predicted values and exhibits much lower variance across different trials.

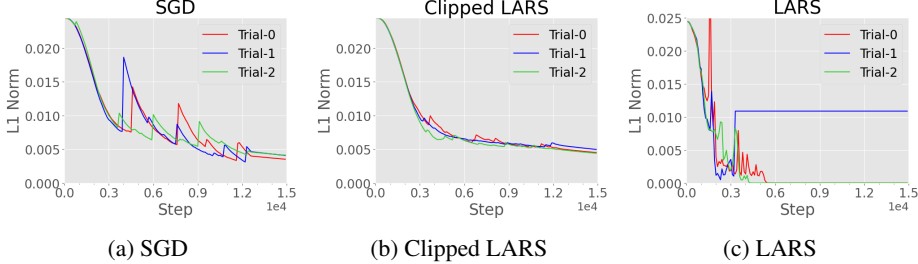

(a) SGD        (b) Clipped LARS        (c) LARS

Figure 3: L1 norms of the parameters of the representation network of several trials with a larger batch size using SGD, Clipped LARS, and LARS respectively in Breakout. (a) There exist several sudden huge changes in the weights across all trials when using SGD. (b) Clipped LARS significantly stabilizes the training process. (c) LARS causes over-regularization in the initial training stage and numerical instability or excessive shrinkage of the gradient during further training.

Distributed Prioritized Experience Replay (DPER) (Horgan et al., 2018) is that DPER only updates priorities of data points that are trained on, while P-Refresh updates priorities of all data points. This difference allows SpeedyZero to effectively use data from old policies, which stabilizes the training and leads to better performance.

**Clipped LARS:** To tackle the unstable issue of large batch size training, we propose an optimizer called Clipped LARS. Clipped LARS updates the parameters with the following rule,

$$w_{t+1} = w_t - \gamma \cdot \min\left(\frac{\eta||w_t||_2}{||\nabla w_t||_2 + \beta||w_t||_2}, 1\right)$$
$$\cdot \left(||\nabla w_t||_2 + \beta||w_t||_2\right) \quad (1)$$

where $w_t$ is the parameter of a layer after $t$ training steps, $\beta$ is the weight decay coefficient, $\gamma$ is the base learning rate, $\eta$ is a scaling factor to control the change in the parameter. As shown in Fig. 3c, LARS (You et al., 2017) shows an over-regularization effect in the early training stage. Clipped LARS overcomes the over-regularization issue by clipping the scaling ratio to less than 1 to avoid magnifying the gradients but only shrinking the exploding gradients. More details about Clipped LARS can be found in Appendix A.3.

### 4.4 IMPLEMENTATION

SpeedyZero is highly optimized for higher training throughput and lower latency on critical modules. In this section, we will introduce some key system efficiency optimizations. For more details about the implementation of SpeedyZero, please refer to the appendix.

**Distributed Data Parallel:** We use Distributed Data Parallel provided by PyTorch (Li et al., 2020) to amortize batch size on multiple GPUs for faster training.

**Data Compression:** We compress the batches sent through the network with the lz4 algorithm, reducing the average network bandwidth requirement by $12\times$. Also, lz4 features high compression and decompression throughput, causing negligible overhead to our overall latency.

**Replay Buffer Replication:** We keep a replica of the replay buffer on each reanalysis node. This grants data loaders fast access to the data they need and also reduces traffic over the network. Each trajectory is only transmitted once while priorities are synchronized throughout the training process.

## 5 EXPERIMENTS

### 5.1 EXPERIMENT SETUP

**Atari 100k Benchmark.** The Atari 100k benchmark is proposed for testing the effectiveness of sample efficient RL methods (Kaiser et al., 2019). It contains 26 Atari games that are deemed solvable with a limited amount of samples. In this benchmark, agents are allowed to take at most 100 thousand environment steps, which are equivalent to 400 thousand frames due to a frameskip of 4. EfficientZero is the first method that achieves human performance in terms of both the mean and median of the human normalized score on this benchmark. In our experiments, we test SpeedyZero on the Atari 100k benchmark and also conduct additional experiments on the same set of games with 300k environment steps. Raw performance on each game as well as the mean and median of the human normalized score is reported. Human normalized score is computed as $(\text{score}_{\text{agent}} - \text{score}_{\text{random}})/(\text{score}_{\text{human}} - \text{score}_{\text{random}})$. The baselines we compare against include SimPLe (Kaiser et al., 2019), CuRL (Srinivas et al., 2020), SPR (Schwarzer et al., 2020), MuZero and EfficientZero. All baselines use 100k environment steps.

For the main results in Sec. 5.2 and ablation study in Sec. 5.3, the trainer node is configured with 8 DDP trainers and each DDP trainer receives batches with batch size 256 for training, indicating a total batch size of 2048. The model held by each Reanalyze workers is updated every 25 training steps. The models of the priority refreshers and actors are updated every 10 training steps. The total number of training steps is 15k. We run SpeedyZero with two different clusters, resulting in 35min and 50min total running time due to differences in the machine hardware. The detailed hardware configuration of the two clusters is listed in the appendix. As a reference, EfficientZero uses a batch size of 256 and 120k training steps, taking over 8.5 hours to finish training under the same machine configuration used by the 35min experiments of SpeedyZero.

In Sec. 5.4, we carry out case studies on large batch size training with limited steps and point out bottlenecks that prevent SpeedyZero from using larger batch sizes and less training time. We compare SpeedyZero with model-free method PPO (Schulman et al., 2017) under the setting of large batch size training on a subset of representative environments. PPO uses 25 million environment steps for training. The full scaling policy for SpeedyZero is given in the appendix.

### 5.2 MASTERING ATARI IN 30 MINUTES

Table. 1 compares the result of SpeedyZero under different cluster configurations and number of environment steps with the result of other methods on the set of games in the Atari 100k benchmark. SpeedyZero surpasses human performance on both the normalized mean score and the normalized median score. Running with 100k environment steps for 50 minutes, SpeedyZero achieves a normalized mean of 1.483 and a normalized median of 1.011. Compared with EfficientZero, which uses 8.5 hours on each game, SpeedyZero achieves 10× speedup. The performance gap is mainly caused by the usage of a larger batch size in our setting.

Additionally, We perform experiments with 300k environment steps on two different clusters which allow SpeedyZero to finish training in 35 and 50 minutes. In the 35 minutes experiment, which accelerates EfficientZero by 14.5×, SpeedyZero achieves a normalized mean of 2.594 and normalized median of 0.520. In the 50 minutes experiment, SpeedyZero achieves a similar normalized mean to the 35 minutes experiment, i.e. 2.915, and shows a stronger normalized median of 1.113. We observe severe performance degradation on some environments, e.g. Jamesbond, when shifting from the 50min experiment to the 35min experiment. The inconsistency of the performance among different machine configurations also occurred in our early experiments. (See A.6 for more details.) The main reason for the performance gap is the non-uniform speedup of different components on machines with faster GPUs. For example, increasing the training speed while not accelerating the data generation process will influence the "priority staleness", which measures the model version gap between when a batch is sampled from the replay buffer and when it is trained on.

### 5.3 ABLATION STUDY

**Priority Refresh.** As stated in Sec. 4.3, the RL agent's final performance suffers a lot from the unstable predicted values during the initial training stage. Therefore we propose Priority Refresh to stabilize the training process. As shown in Fig. 2, when using uniform sampling, the predicted values

| Game | Random | Human | SimPLE | CuRL | SPR | MuZero | EfficientZero | SpeedyZero | | |
|---|---|---|---|---|---|---|---|---|---|---|
| Environment Steps | / | / | 100k | 100k | 100k | 100k | 100k | 300k | 300k | 100k |
| Time | / | / | / | / | / | 8.5 hours | 8.5 hours | **35min** | **50min** | **50min** |
| Alien | 227.8 | 7127.7 | 616.9 | 558.2 | 801.5 | 530.0 | **1140.3** | 627.3 | 1058.3 | 718.0 |
| Amidar | 5.8 | 1719.5 | 88.0 | 142.1 | **176.3** | 38.8 | 101.9 | 85.8 | 153.5 | 86.1 |
| Assault | 222.4 | 742.0 | 527.2 | 600.6 | 571.0 | 500.1 | **1407.3** | 1241.1 | 1129.0 | 952.5 |
| Asterix | 210.0 | 8503.3 | 1128.3 | 734.5 | 977.8 | 1734.0 | 16843.8 | 124148.3 | **177142.9** | 11019.3 |
| Bank Heist | 14.2 | 753.1 | 34.2 | 131.6 | **380.9** | 192.5 | 361.9 | 166.7 | 239.5 | 228.6 |
| BattleZone | 2360.0 | 37187.5 | 5184.4 | 14870.0 | 16651.0 | 7687.5 | **17938.0** | 10873.3 | 9930.0 | 7437.5 |
| Boxing | 0.1 | 12.1 | 9.1 | 1.2 | 35.8 | 15.1 | **44.1** | 33.7 | 28.1 | 29.0 |
| Breakout | 1.7 | 30.5 | 16.4 | 4.9 | 17.1 | 48.0 | **406.5** | 401.1 | 404.1 | 371.1 |
| ChopperCmd | 811.0 | 7387.8 | 1246.9 | 1058.5 | 974.8 | 1350.0 | **1794.0** | 935.0 | 1168.0 | 1254.8 |
| CrazyClimber | 10780.5 | 35829.4 | 62583.6 | 12146.5 | 42923.6 | 56937.0 | 80125.3 | **113582.7** | 107975.4 | 82106.3 |
| Demon Attack | 152.1 | 1971.0 | 208.1 | 817.6 | 545.2 | 3527.0 | 13298.0 | 25603.3 | **26023.3** | 9097.7 |
| Freeway | 0.0 | 29.6 | 20.3 | **26.7** | 24.4 | 21.8 | 21.8 | 0.0 | 10.7 | 7.1 |
| Frostbite | 65.2 | 4334.7 | 254.7 | 1181.3 | **1821.5** | 255.0 | 313.8 | 406.5 | 831.4 | 262.6 |
| Gopher | 257.6 | 2412.5 | 771.0 | 669.3 | 715.2 | 1256.0 | 3518.5 | 3907.9 | **4108.2** | 2600.3 |
| Hero | 1027 | 30826.4 | 2656.6 | 6279.3 | 7019.2 | 3095.0 | **8530.1** | 7601.9 | 5650.9 | 6002.5 |
| Jamesbond | 29.0 | 302.8 | 125.3 | 471.0 | 365.4 | 87.5 | **459.4** | 169.5 | 342.3 | 340.1 |
| Kangaroo | 52.0 | 3035.0 | 323.1 | 872.5 | **3276.4** | 62.5 | 962.0 | 1403.3 | 542.5 | 428.7 |
| Krull | 1598.0 | 2665.5 | 4539.9 | 4229.6 | 3688.9 | 4890.8 | 6047.0 | **7677.3** | 6388.3 | 5855.15 |
| Kung Fu Master | 258.5 | 22736.3 | 17257.2 | 14307.8 | 13192.7 | 18813.0 | **31112.5** | 30964.7 | 29131.5 | 22170.2 |
| Ms Pacman | 307.3 | 6951.6 | 1480.0 | 1465.5 | 1313.2 | 1265.6 | 1387.0 | 2685.0 | **3296.1** | 1586.6 |
| Pong | -20.7 | 14.6 | 12.8 | -16.5 | -5.9 | -6.7 | **20.6** | 15.0 | 17.5 | 16.3 |
| Private Eye | 24.9 | 69571.3 | 58.3 | **218.4** | 124.0 | 56.3 | 100.0 | 0.0 | 33.3 | 29.7 |
| Qbert | 163.9 | 13455.0 | 1288.8 | 1042.4 | 669.1 | 3952.0 | 15458.1 | 14205.0 | **16043.1** | 14467.0 |
| Road Runner | 11.5 | 7845.0 | 5640.6 | 5661.0 | 14220.5 | 2500.0 | 18512.5 | 17235.0 | **25368.0** | 8496.3 |
| Seaquest | 68.4 | 42054.7 | 683.3 | 384.5 | 583.1 | 208.0 | 1020.5 | 901.6 | **1312.2** | 530.7 |
| Up N Down | 533.4 | 11693.2 | 3350.3 | 2955.2 | **28138.5** | 2896.9 | 16095.7 | 6411.0 | 22531.8 | 12383.1 |
| Normed Mean | 0.000 | 1.000 | 0.443 | 0.381 | 0.704 | 0.562 | 1.904 | 2.594 | **2.915** | 1.483 |
| Normed Median | 0.000 | 1.000 | 0.144 | 0.175 | 0.415 | 0.227 | **1.160** | 0.520 | 1.113 | 1.011 |

Table 1: Scores and running time achieved by SpeedyZero and some baselines on the Atari 100k benchmark. Compared with previous RL methods, SpeedyZero achieves human-level performance and performs best on 9 out of 26 games with $10\times$ shorter training time. The results of SpeedyZero are evaluated with 100 evaluation episodes. The 35min results of SpeedyZero with 300k environment steps are evaluated with 3 training seeds. The 50min results of SpeedyZero are evaluated with 16 training seeds. The training time of EfficientZero is evaluated under the same machine configuration of the 300k, 35 minutes experiment of SpeedyZero.

| Game | DPER | Uniform | P-Refresh |
|---|---|---|---|
| Pong | 16.0 | 13.0 | **17.5** |
| Jamesbond | 284.2 | 251.3 | **342.3** |
| UpNDown | 21634.3 | 20432.3 | **22531.8** |

Table 2: Ablation study on different training data sampling strategy, i.e., DPER, uniform sampling, and Priority Refresh. Unif. is the worst in all environments while Priority Refresh outperforms all the baselines in these environments.

remain unstable throughout the training process. When using DPER, although the values sometimes seem more stable, the variance across different trials is still considerably high. In contrast, Priority Refresh ensures stable training within a single run and across multiple runs. In Table. 2, Priority Refresh achieves the highest score among all sampling methods in a set of environments. This suggests the superiority of Priority Refresh in stabilizing training and improving final performance.

**Clipped LARS.** Table. 3 compares the performance of SpeedyZero when using SGD, LARS, and Clipped LARS as the optimizer. When using LARS, training fails completely. Clipped LARS stabilizes large batch size training of SpeedyZero and significantly improves the performance over SGD and LARS, indicating that Clipped LARS is critical to the overall success of SpeedyZero.

## 5.4 EFFECT OF BATCH SIZE

Prior works have shown that PPO can be easily parallelized and benefit from large batch size training (Stooke & Abbeel, 2018). However, we find that it is hard to train with larger batch sizes in

| Optimizer | SGD | LARS | Clipped LARS |
|---|---|---|---|
| Asterix | 2732.0 | 200.0 | **11019.3** |
| Breakout | 88.7 | 0.6 | **371.1** |
| Gopher | 1459.4 | 0.6 | **2600.3** |
| Pong | 2.1 | -19.9 | **16.3** |

Table 3: Performance of SpeedyZero using SGD, LARS, and Clipped LARS on a number of selected Atari games with a batch size of 2048. LARS completely fails to learn any useful policies. Clipped LARS is significantly better than SGD and LARS.

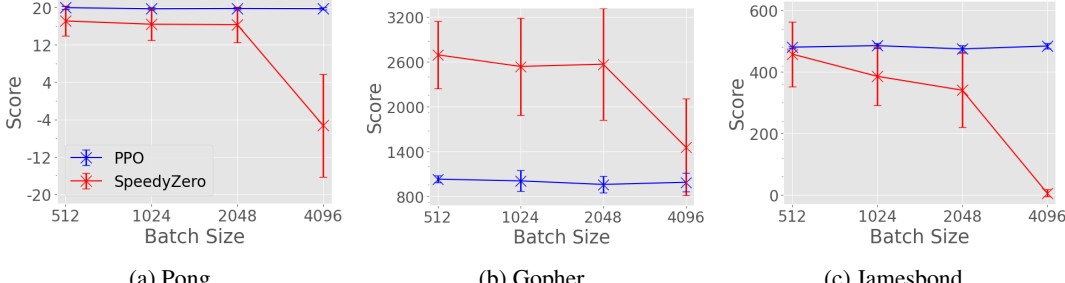

|  | (a) Pong |  | (b) Gopher |  | (c) Jamesbond |

Figure 4: The effect of batch size for SpeedyZero comparing with a distributed implementation of PPO. We report the scores of PPO and SpeedyZero on selected games with batch size of 512/1024/2048/4096. Here PPO consumes 25M samples while SpeedyZero uses 100k samples. With an increasing batch size, PPO maintains its performance while the performance of SpeedyZero drops a lot. The most significant drop happens when increasing the batch size from 2048 to 4096.

| Batch Size | Oracle Staleness | SpeedyZero Staleness | Staleness Ratio |
|---|---|---|---|
| 512 | 50 | 50.8 | 1.016 |
| 1024 | 25 | 36.3 | 1.452 |
| 2048 | 12.5 | 37.2 | 2.976 |

Table 4: Bottleneck analysis on the Reanalyze staleness. Oracle staleness is the optimal staleness that could be achieved by the synchronized version of SpeedyZero. SpeedyZero Staleness measures the actual reanalyze staleness achieved by SpeedyZero. Finally, "Staleness Ratio" is the ratio of SpeedyZero staleness to the oracle staleness. A larger staleness ratio indicates a larger gap between the synchronized and asynchronized execution. As the batch size increases, the reanalyze ratio becomes larger, indicating a much more severe gap between SpeedyZero and the synchronized version.

SpeedyZero. Fig. 4 compares PPO and SpeedyZero when using different batch sizes in a number of selected environments. PPO doesn't show performance degradation and maintains its performance when using larger batch size. However, SpeedyZero demonstrates different degrees of performance degradation in different environments and shows the most significant performance degradation when increasing the batch size from 2048 to 4096. We find the "Reanalyze staleness" as a critical factor influencing the performance of SpeedyZero and blocking the learning of SpeedyZero.

**Reanalyze Staleness.** Since reanalysis and training are parallelized, DDP trainers often do not receive batches that are reanalyzed by the latest target model but by an old version of the model. This model version gap of the training batches, which we call "Reanalyze staleness", could significantly affect the quality of training. In practice, we find that there could be several reasons contributing to an increased Reanalyze staleness, including improper queue design between the reanalysis node and the trainer node, a large latency of the Reanalyze processes, communication overhead and compression time over the training batches.

When using a larger batch size, which requires a shorter interval to update the target model, the issue of Reanalyze staleness becomes more severe since the latency of components except for the trainers remains the same while the interval between two consecutive target model updates is much shorter. Table. 4 reports the ratio of actual SpeedyZero Reanalyze staleness to the oracle Reanalyze staleness when using different batch sizes. As expected, the ratio increases as the batch size increases, indicating a much more severe gap between SpeedyZero and the synchronized version.

## 6 CONCLUSION

In this work, we have developed a fast and sample efficient model-based RL training system, SpeedyZero, and introduced a new priority experience replay method, Priority Refresh, and an optimizer, Clipped LARS, to stabilize training. With the highly optimized system design and algorithmic improvements combined, SpeedyZero achieves human-level performance on the Atari 100k benchmark with only 300k samples and 35 minutes of training. This work is one step towards the application of RL in real-world scenarios where both sample efficiency and training time are mission-critical. We expect future research to further accelerate SpeedyZero and apply SpeedyZero in the real world.

ACKNOWLEDGMENTS

This work is supported by the Ministry of Science and Technology of the People´s Republic of China, the 2030 Innovation Megaprojects "Program on New Generation Artificial Intelligence" (Grant No. 2021AAA0150000).

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

## A APPENDIX

### A.1 CLUSTER HARDWARE CONFIGURATIONS

This section lists the cluster hardware configurations of the 35min and 50min experiments. For the 35min experiments, the trainer node and the data node are both machines with 8 A100 80G GPUs (with NV-Switch), 128 CPU cores, and 1TB of RAM. There are 9 reanalysis nodes, each of which contains 4 A100 80G GPUs (with NV-Switch), 64 CPU cores, and 512GB of RAM. For the 50min experiments, the trainer node and the data node both contain 8 A100 80G GPUs (without NV-Switch), 128 CPU cores, and 512GB of RAM and the 15 reanalysis nodes all contain 1 NVIDIA RTX 3090 GPUs, 128 CPU cores, and 512GB of RAM.

### A.2 MODELS AND HYPER-PARAMETERS

SpeedyZero uses the same model as EfficientZero. The model consists of three modules: representation function, dynamics function, and prediction function, which are all represented as neural networks. We list the architecture of each module below:

| Layer Type | Configuration | Activation | Batch Norm |
|---|---|---|---|
| **Representation Network** | | | |
| Convolution | kernel 3x3, stride 2, 32 output planes, output resolution 48×48 | ReLU | Yes |
| Residual Block | 32 planes | - | - |
| Residual Downsample Block | stride 2, 64 output planes, output resolution 24×24 | - | - |
| Residual Block | 64 planes | - | - |
| Average Pooling | stride 2, output resolution 12×12 | ReLU | Yes |
| Residual Block | 64 planes | - | - |
| Average Pooling | stride 2, output resolution 6×6 | ReLU | Yes |
| Residual Block | 64 planes | - | - |
| **Dynamics Network** | | | |
| - | Concate the latent states with input actions into 65 planes | - | - |
| Convolution | kernel 3x3, stride 2, 64 output planes | - | Yes |
| Convolution | kernel 3x3, stride 2, 32 output planes, output resolution 48×48 | - | Yes |
| - | A residual link adds the input latent states with output | ReLU | No |
| Residual Block | 64 planes | - | - |
| **Prediction Network** | | | |
| Residual Block | 64 planes | - | - |
| Convolution | kernel 1x1, 16 output planes | ReLU | Yes |
| - | Flatten | - | - |
| Linear | D output dimensions (D differs for reward/policy/value) | - | - |
| **Value-prefix Prediction Network** | | | |
| Convolution | kernel 1x1, 16 output planes | ReLU | Yes |
| - | Flatten | - | - |
| LSTM | 512 hidden size | ReLU | Yes |
| Linear | 32 output dims | ReLU | Yes |
| Linear | 601 output dims | - | - |

Table 5: Network architecture of SpeedyZero.

We stack four historical frames as the input observations, with an interval of 4 frame-skip. The frames are staked on the channel dimension, hence the shape of the input is $96 \times 96 \times 12$.

Common hyper-parameters of SpeedyZero are as shown in Table. 6. The search budget is 50 in the 50min experiments while it is decreased to 45 in the 35min experiments to speedy up the generation of interactions with the environment.

### A.3 CLIPPED LARS

In practice, we use momentum SGD during the initial training process and then switch to Clipped LARS after a number of training steps. When applying Clipped LARS, we use $\eta = 0.05$ for batch normalization layers and biases in all layers. For other layers, a smaller $\eta$ is used and is specifically tuned for different games. Since we follow the initialization scheme of EfficientZero to set the last linear layers of the value, policy, and reward prediction networks to zero, Clipped LARS is only activated after the norms of these layers are above a constant threshold.

| Parameter Name | Value |
|---|---|
| Observation down-sampling | $96 \times 96$ |
| Frame staked | 4 |
| Frame skip | 4 |
| Reward clipping | True |
| Terminal on loss of life | True |
| Max frames per episode | 12K |
| Discount factor | $0.997^4$ |
| Optimizer | SGD |
| Max gradient norm | 10 |
| Priority exponent ($\alpha$) | 0.6 |
| Evaluation episodes | 100 |
| Actor model update interval | 10 |
| Unroll steps | 5 |
| TD steps | 5 |
| Policy loss coefficient | 1 |
| Value loss coefficient | 0.25 |
| Self-supervised consistency loss coefficient | 2 |
| LSTM horizon length | 5 |
| Dirichlet noise ratio | 0.3 |
| Number of simulations in MCTS | 45/50 |

Table 6: Common hyper-parameters of SpeedyZero.

| Game | $\eta$ |
|---|---|
| Asterix | 0.01 |
| Gopher | 0.01 |
| KungFuMaster | 0.005 |
| Pong | 0.005 |
| RoadRunner | 0.005 |
| UpNDown | 0.005 |

Table 7: Best $\eta$ found for a number of selected games. For other games, we use the default $\eta = 0.03$.

We specifically perform grid search for $\eta$ from $[0.03, 0.01, 0.005, 0.001]$ on Asterix, Gopher, Kung-FuMaster, Pong, RoadRunner, and UpNDown. The best $\eta$ found for the selected games are listed in Table. 7. For other games, we use $\eta = 0.03$ by default.

## A.4 Large Batch Size Experiment Setup

For SpeedyZero, We use the linear scaling rule for large batch size training. The hyper-parameters for large batch size training are shown in Table. 8.

| Batch Size | Target Model Update Interval | Training Steps | Learning Rate | SGD Init Steps |
|---|---|---|---|---|
| 512 | 100 | 60k | $\frac{8}{10 \times 2^{1.5}}$ | 6k |
| 1024 | 50 | 30k | $\frac{8}{10 \times 2^{1}}$ | 3k |
| 2048 | 25 | 15k | $\frac{8}{10 \times 2^{0.5}}$ | 600 |

Table 8: Hyper-parameters of SpeedyZero with different batch sizes.

For PPO, we only change the batch size for different experiments. The hyper-parameters of PPO are shown in Table. 9. The network architecture we use for PPO is shown in Table. 10.

| Parameter Name | Value |
|---|---|
| Observation down-sampling | $84 \times 84$ |
| Frame staked | 4 |
| Frame skip | 4 |
| Learning rate | 5e-5 |
| Discount | 0.99 |
| Env steps | 25M |
| PPO Epoch | 4 |
| GAE parameter ($\lambda$) | 0.95 |
| Number of actors | 8 |
| Entropy coeff. | 0.01 |
| VF coeff. | 0.5 |

Table 9: Hyper-parameters of PPO

| Layer Type | Configuration | Activation | Batch Norm |
|---|---|---|---|
| Convolution | kernel 8x8, stride 4, 32 output planes | ReLU | - |
| Convolution | kernel 4x4, stride 2, 64 output planes | ReLU | - |
| Convolution | kernel 3x3, stride 1, 64 output planes | ReLU | - |
| Linear | 512 ouput dims | ReLU | - |
| Linear | $action_{dim}$ output dims | - | - |

Table 10: Network architecture of PPO.

## A.5 SYSTEM WORKFLOW AND SYSTEM CONFIGURATIONS

In this section, we describe the detailed workflow of SpeedyZero. We illustrate this by looking at how one data point (environment step) takes effect in the whole training process. The story begins on the data node. Actors on the data node collect the data point from the environment and put it into the replay buffer. Priorities refreshers periodically recompute the priorities of data points in the replay buffer. Since we have replay buffer replicas on reanalysis nodes, we need to synchronize the data points and their priorities to the replicas. Each data point is only sent once (when the trajectory is finished) while priorities are synchronized periodically throughout the training process.

On the reanalysis node, data loaders sample batches based on previously computed priorities and reformat the batches for ease of GPU reanalysis. The batches are then sent into a shared memory queue between data loaders and Reanalyze workers. The separation of data loading and reanalyzing decouples CPU workload with GPU workload, improving resource utilization. During our implementation, we partition data loaders and Reanalyze workers into several groups. Each group has one shared memory queue, and the data loaders can only communicate with Reanalyze workers in the same group. We use grouping here to achieve higher bandwidth and lower latency. Continue with the workflow, Reanalyze workers will then pop batches from the shared memory queue and reanalyze them using MCTS. The Reanalyze workers will not send the reanalyzed batches to the trainers. Instead, they will push the batches into another shared memory queue, called the batch queue. Batch senders will then take out the batches from the batch queue and send the batches through the network to the trainer node. This additional step overlaps slow network transmission with computation since Reanalyze workers can work on the next batch when the batch senders are sending the current batch.

Similarly, on the trainer node, the DDP trainers do not receive batches themselves. The batch receivers are responsible for batch receiving and they will put the batches into a trainer-side batch queue. The DDP trainers directly read the batches from this queue and use the batches in training.

In Table. 11, we show some key system configurations used to setup SpeedyZero on our cluster. This can serve as a reference when setting up SpeedyZero on new clusters.

| Configuration Name | Value |
|---|:---:|
| **Data Node** | |
| Number of Actors | 8 |
| Number of Priority Refreshers | 3 |
| Replay Buffer Capacity (in number of trajectories) | 8192 |
| Replay Buffer Capacity (in memory consumption) | ˜32GB |
| **Reanalysis Node** | |
| Number of Data Loaders | 36 |
| Number of Reanalyze workers | 24 |
| Number of Batch Senders | 16 |
| Number of Data Loader - Reanalyer Queues | 8 |
| Data Loader - Reanalyer Queues Capacity (in number of batches, each) | 32 |
| Data Loader - Reanalyer Queues Capacity (in memory consumption, each) | ˜32GB |
| Number of Batch Storages | 8 |
| Batch Storage Capacity (in number of batches, each) | 16 |
| Batch Storage Capacity (in memory consumption, each) | ˜8GB |
| **Trainer Node** | |
| Number of DDP Trainers | 8 |
| Number of Batch Receivers | 8 |
| Number of Batch Storages | 1 |
| Batch Storage Capacity (in number of batches, each) | 64 |
| Batch Storage Capacity (in memory consumption, each) | ˜16GB |
| Signal Queue Capacity (in number of training signals) | 16 |
| Signal Queue Capacity (in memory consumption) | ˜2GB |

Table 11: Key system configuration of SpeedyZero.

## A.6 EARLY EXPERIMENT RESULTS

In our early experiments, we tested SpeedyZero with a smaller batch size, i.e. 512, with a significantly decreased number of training steps, i.e. 20k. We use a small target model update interval of 40. The experiment results are shown in Table. 12. The performance shows a large variance across different machine configurations. The main reason for the performance gap between these experiments is the non-uniform speedup of different components on machines with faster GPUs. For example, increasing the training speed while not accelerating the data generating process will influence the "priority staleness", which measures the model version gap between when a batch is sampled from the replay buffer and when it is trained on. It is an interesting and challenging direction to stabilize the performance of RL agents across different hardware configurations.

**Priority Staleness.** As data loaders run in parallel with the DDP trainers, there is a time gap between a batch is sampled and is trained on. This means that datapoints in the batch are sampled according to priorities from an old version of the model. This gap is measured by "priority staleness", i.e. the step when the batch is trained on minus the step when it is sampled. We ablate different degrees of priority staleness in Table. 13. When the priority staleness is very small, SpeedyZero has poor performance. We hypothesize that the reason is that a larger priority staleness brings some regularization effect on the sampling probability, hence preventing the training from only focusing on a limited number of datapoints. However, the priority staleness is hard to control in SpeedyZero, since it depends on the latency of multiple components in the system, which could differ a lot on machines with different configurations. We leave the study of optimal priority staleness and improved schemes to control the priority staleness as future work.

| Staleness | 15 | 20 | 35 |
|---|:---:|:---:|:---:|
| Pong | -0.7 | 1.8 | **18.2** |
| Jamesbond | 351.3 | 494.3 | **535.5** |
| UpNDown | 4268.4 | 4582.6 | **10064** |

Table 13: Performance in Pong, Jamesbond and UpNDown with different degrees of priority staleness. The result indicates that the priority staleness should not be too small, i.e., the priorities should be recomputed at a proper frequency, for the best performance.

| Game | Random | Human | SimPLE | CuRL | SPR | MuZero | EfficientZero | SpeedyZero | | | |
|---|---|---|---|---|---|---|---|---|---|---|---|
| Environment Steps | / | / | 100k | 100k | 100k | 100k | 100k | 300k | 300k | 300k | 100k |
| Time | / | / | / | / | / | 8.5 hours | 8.5 hours | 0.5 hours | 0.75 hours | 1 hour | 0.75hours |
| Alien | 227.8 | 7127.7 | 616.9 | 558.2 | 801.5 | 530.0 | 1140.3 | 818.4 | 734.1 | 1177.7 | 543.2 |
| Amidar | 5.8 | 1719.5 | 88.0 | 142.1 | 176.3 | 38.8 | 101.9 | 73.5 | 98.8 | 112.7 | 68.7 |
| Assault | 222.4 | 742.0 | 527.2 | 600.6 | 571.0 | 500.1 | 1407.3 | 991.5 | 906.2 | 1470.0 | 616.4 |
| Asterix | 210.0 | 8503.3 | 1128.3 | 734.5 | 977.8 | 1734.0 | 16843.8 | 41993.3 | 67179.1 | 49494.6 | 4226.9 |
| Bank Heist | 14.2 | 753.1 | 34.2 | 131.6 | 380.9 | 192.5 | 361.9 | 205.8 | 217.6 | 187.3 | 128.8 |
| BattleZone | 2360.0 | 37187.5 | 5184.4 | 14870.0 | 16651.0 | 7687.5 | 17938.0 | 9655.3 | 6418.1 | 8906.3 | 5861 |
| Boxing | 0.1 | 12.1 | 9.1 | 1.2 | 35.8 | 15.1 | 44.1 | 21.2 | 31.3 | 47.0 | 15.5 |
| Breakout | 1.7 | 30.5 | 16.4 | 4.9 | 17.1 | 48.0 | 406.5 | 258.6 | 340.9 | 409.8 | 304.6 |
| ChopperCmd | 811.0 | 7387.8 | 1246.9 | 1058.5 | 974.8 | 1350.0 | 1794.0 | 356.7 | 472.1 | 453 | 408.8 |
| CrazyClimber | 10780.5 | 35829.4 | 62583.6 | 12146.5 | 42923.6 | 56937.0 | 80125.3 | 83232.3 | 98454.9 | 93455.2 | 69929.4 |
| Demon Attack | 152.1 | 1971.0 | 208.1 | 817.6 | 545.2 | 3527.0 | 13298.0 | 54994.4 | 34404.9 | 39699.7 | 5407.9 |
| Freeway | 0.0 | 29.6 | 20.3 | 26.7 | 24.4 | 21.8 | 21.8 | 0.0 | 4.0 | 0.0 | 13.9 |
| Frostbite | 65.2 | 4334.7 | 254.7 | 1181.3 | 1821.5 | 255.0 | 313.8 | 1105.2 | 561.7 | 238.9 | 261.1 |
| Gopher | 257.6 | 2412.5 | 771.0 | 669.3 | 715.2 | 1256.0 | 3518.5 | 2465.5 | 1989.7 | 2929.4 | 1481.4 |
| Hero | 1027 | 30826.4 | 2656.6 | 6279.3 | 7019.2 | 3095.0 | 8530.1 | 7806.1 | 7633.0 | 8461.3 | 6115.2 |
| Jamesbond | 29.0 | 302.8 | 125.3 | 471.0 | 365.4 | 87.5 | 459.4 | 494.3 | 351.6 | 535.5 | 289.9 |
| Kangaroo | 52.0 | 3035.0 | 323.1 | 872.5 | 3276.4 | 62.5 | 962.0 | 411.3 | 1012.7 | 1246.9 | 525.9 |
| Krull | 1598.0 | 2665.5 | 4539.9 | 4229.6 | 3688.9 | 4890.8 | 6047.0 | 6771.6 | 5659.6 | 7962.6 | 5470.5 |
| Kung Fu Master | 258.5 | 22736.3 | 17257.2 | 14307.8 | 13192.7 | 18813.0 | 31112.5 | 24159.7 | 19958.8 | 21700 | 17253.5 |
| Ms Pacman | 307.3 | 6951.6 | 1480.0 | 1465.5 | 1313.2 | 1265.6 | 1387.0 | 1152.5 | 1716.6 | 3386.8 | 805.5 |
| Pong | -20.7 | 14.6 | 12.8 | -16.5 | -5.9 | -6.7 | 20.6 | 1.8 | 3.8 | 18.2 | -4.9 |
| Private Eye | 24.9 | 69571.3 | 58.3 | 218.4 | 124.0 | 56.3 | 100.0 | 75.7 | 55.3 | 48.5 | 63 |
| Qbert | 163.9 | 13455.0 | 1288.8 | 1042.4 | 669.1 | 3952.0 | 15458.1 | 9584.7 | 9034.4 | 16334.2 | 4389 |
| Road Runner | 11.5 | 7845.0 | 5640.6 | 5661.0 | 14220.5 | 2500.0 | 18512.5 | 13350.0 | 11942.1 | 21692.2 | 5053.3 |
| Seaquest | 68.4 | 42054.7 | 683.3 | 384.5 | 583.1 | 208.0 | 1020.5 | 494.4 | 663.7 | 1047.5 | 367.1 |
| Up N Down | 533.4 | 11693.2 | 3350.3 | 2955.2 | 28138.5 | 2896.9 | 16095.7 | 4582.6 | 4942.7 | 10064 | 2376.1 |
| Normed Mean | 0.000 | 1.000 | 0.443 | 0.381 | 0.704 | 0.562 | 1.904 | 2.440 | 2.213 | 2.660 | 1.038 |
| Normed Median | 0.000 | 1.000 | 0.144 | 0.175 | 0.415 | 0.227 | 1.160 | 0.500 | 0.531 | 0.904 | 0.383 |

Table 12: Early experiment results with a batch size of 512 and 20k training steps. Compared with previous RL methods, SpeedyZero achieves human-level performance and performs best on 12 out of 26 games with $10\times$ shorter training time. The results of SpeedyZero are evaluated with 100 evaluation episodes. The 0.5 hour and 1 hour results of SpeedyZero with 300k environment steps are evaluated with 3 training seeds. The 0.75 hour results of SpeedyZero with 100k and 300k environment steps are evaluated with 16 seeds. The training time of EfficientZero is evaluated under the same machine configuration of the 300k, 30 minutes experiment of SpeedyZero.

