# OpenReview forum: "SpeedyZero: Mastering Atari with Limited Data and Time"
_ICLR.cc/2023/Conference — ICLR 2023 poster_

### Official Review · Reviewer_Rw6q · 2022-10-24

**Confidence:** 3
**Correctness:** 2
**Technical Novelty And Significance:** 2
**Empirical Novelty And Significance:** 2
**Recommendation:** 5

**Clarity, Quality, Novelty And Reproducibility:**

The paper appears fairly clear despite aforementioned proofreading issues, and I was easily able to identify the novelty and understand the methodology. Given the authors did not include code, the reproducibility is questionable, and I implore the authors to release the code when ready.

**Strength And Weaknesses:**

Strengths:
* The overall idea appears good, and creating asynchronous nodes makes sense when constructing effective compute strategies for model-based/off-policy approaches.
* The proposed P-Refresh algorithm is simple but appears important in the high-throughput problem setting that the authors consider.

Weaknesses:
* I have several concerns with the experimental methodology of this work.
    * The benchmarking does not appear to make sense; it would appear the only difference between the 30 and 60m SpeedyZero agents is the type of compute node used, yet the results (specifically the Median) appear vastly different. I would be willing to accept a difference if there was also a change in hyperparameters (for instance fewer rollout steps), but it appears the same algorithm was deployed in each case. This high variance is a concern as it is unclear how strong the performance really is. Weirdly enough, the authors appear to attribute the stronger performance of the 60m run to the slower machine: *"...if SpeedyZero is allowed to run for 1 hour, it performs even better..."*.
    * Similarly, the EfficientZero algorithm is run on a different node altogether, which makes direct comparison hard. To my understanding, A100 GPUs run faster than the 3090 GPUs used on the EfficientZero cluster. As a higher note, it makes no sense that these algorithms were run on vastly different hardware, specifically given a key comparison is in their run times, so controlling for this would appear vital.
    * Related to the initial point, the number of seeds seems very low. This may explain the extreme variance that we see across the 30 and 60m runs.
    * It seems unfair to compare to other approaches when using 3x the data (300k v.s. 100k). A key factor behind using model-based approaches is precisely the sample-efficiency gains. I could buy an approach which aims to improve throughput for the purposes of real-life robotics (such as [1]) but this is not the case here. On a related note statements made are simply false: *"SpeedyZero breaks the record on 12 out of 26 games in the Atari 100k benchmark"*; this is clearly factually incorrect given the use of 300k steps.
        * Given the priority on run-time, it therefore makes sense to compare performance at 300k steps against model-free approaches, which may be less sample-efficient, but should run faster than model-based methods. For instance, [2,3] should have publicly available code.
        * In the interests of fairness, SpeedyZero with 100k datapoints should also be compared. This should run very fast (10 mins), and it may be possible to increase compute (e.g., number of updates) such that, while slower than the algorithm presented in the paper, may recover adequate performance.
    * There are some issues with the language, and the authors should proofread their work again. Ones that I spotted:
        * Notably, this phenomenon persists no matter we use distributed... -> Notably, this phenomenon exists when using either DPER or uniform sampling...
        * We hypothesis -> We hypothesise
        * paralleled -> parallelized
        * reanalyze -> Reanalyze (since it is an algorithm); reanalyzers -> Reanalyze workers
        * our model is tiny -> our model is small enough to fit onto a single GPU
        * This suggests the superior -> This suggests the superiority

[1] A Walk in the Park: Learning to Walk in 20 Minutes With Model-Free Reinforcement Learning, Smith et al 2022, arXiv:2208.07860

[2] Image Augmentation Is All You Need: Regularizing Deep Reinforcement Learning from Pixels, Yarats et al, ICLR21

[3] Stabilizing Off-Policy Deep Reinforcement Learning from Pixels, Cetin et al, ICML22


**Summary Of The Paper:**

In this work, the authors improve the time efficiency of EfficientZero by dividing various computational elements across asynchronous nodes. Concretely, by splitting compute tasks (i.e., PER refreshing, Reanalyze/Rollouts, Gradient calcs) across different nodes which communicate asynchronously, the authors are able to massively increase overall data throughput.

This introduces issues with respect to data/priority staleness causing unstable learning, resulting in the introduction of Priority Refresh workers, which act to continuously refresh the priority weights of all data points in the buffer. They show this is addresses the issues induced by stale weights, and reduces variance across seeds.

The authors then show that with a given time budget, their method significantly out performs prior work on the Atari ALE benchmark. Some ablations are then presented showing the impact of removing/changing various components (such as Priority Refresh, batch size).

**Summary Of The Review:**

Overall the paper presents an appealing idea, which is to improve the throughput of model-based methods, and shows promising results on the Atari benchmark. However, there are significant experimental shortcomings that I've listed above. Accordingly, in the event these are not addressed, I cannot recommend accepting the paper in its current form.

---

Reviewers addressed some issues regarding experimental completeness, but questions still remain for me around reproducibility (as reflected in the commments).

---

> ### Author Response · Authors · 2022-11-15
> **Thanks for the review. We have updated the experiments to address the concerns with the experimental methodology and released the code for better reproducibility.**
>
> We thank the reviewer for the constructive review and feedback. We have added the following experiment results to address your concerns:
>
> * SpeedyZero with 100k environment steps (16 seeds).
> * SpeedyZero with 300k environment steps (16 seeds).
> * The training time of EfficientZero on a machine with 8 x A100.
>
> For better reproducibility, we have also released all the code needed to reproduce our results in SpeedyZero in the supplementary material. Here are our detailed responses.
>
> * **Question 1:** The benchmarking does not appear to make sense; it would appear the only difference between the 30 and 60m SpeedyZero agents is the type of compute node used, yet the results (specifically the Median) appear vastly different. I would be willing to accept a difference if there was also a change in hyperparameters (for instance fewer rollout steps), but it appears the same algorithm was deployed in each case. This high variance is a concern as it is unclear how strong the performance really is. Weirdly enough, the authors appear to attribute the stronger performance of the 60m run to the slower machine: "...if SpeedyZero is allowed to run for 1 hour, it performs even better...".
>
> We apologize for the confusing results and analysis. Since the 30 minutes experiments and 60 minutes experiments are performed under different machine configurations, the latency in different components of the system also differs. Simply accelerating the training speed in the trainer node with faster GPU computation is insufficient to ensure a good final performance since it is hard to ensure other system components reach the same speed-up. This non-uniform speedup of different components impairs the RL agent’s performance by disturbing the relative update intervals of different components. This is also the reason that we spend lots of effort optimizing the speed of all system components to alleviate the problem of non-uniform speedup. For example, increasing the training speed while not accelerating the data-generating process will influence the "priority staleness", which measures the model version gap between when a batch is sampled from the replay buffer and when it is trained on. As shown in the ablation study, the priority staleness significantly affects the final performance. We believe that ensuring a stable performance across different hardware setups is a fundamental challenge in accelerating RL methods with distributed computation. We leave the study of the impacts of latency in the different components as future work. We also have updated our manuscript for a clearer description.
>
> * **Question 2:** Similarly, the EfficientZero algorithm is run on a different node altogether, which makes direct comparison hard. To my understanding, A100 GPUs run faster than the 3090 GPUs used on the EfficientZero cluster. As a higher note, it makes no sense that these algorithms were run on vastly different hardware, specifically given a key comparison is in their run times, so controlling for this would appear vital.
>
> We have rerun EfficientZero on a machine with 8xA100 for a fair comparison with SpeedyZero. The time used by EfficientZero is 8.5 hours. On the basis of EfficientZero, SpeedyZero achieves a 17x speedup.
>
> * **Question 3:** Related to the initial point, the number of seeds seems very low. This may explain the extreme variance that we see across the 30 and 60m runs.
>
> We have updated our experiment results with 16 random seeds. As shown in Table. 1, SpeedyZero achieves a mean score of 2.213 and a median score of 0.531 when using 300k samples. For more details about the experiment results, please refer to the paper.
>
> * **Question 4:** It seems unfair to compare to other approaches when using 3x the data (300k v.s. 100k). A key factor behind using model-based approaches is precisely the sample-efficiency gains. I could buy an approach which aims to improve throughput for the purposes of real-life robotics (such as [1]) but this is not the case here. On a related note statements made are simply false: "SpeedyZero breaks the record on 12 out of 26 games in the Atari 100k benchmark"; this is clearly factually incorrect given the use of 300k steps.
>
> We have updated our experiments with experiments using only 100k environment steps, which makes a fair comparison with other methods on the Atari 100k benchmark. When using 100k data, SpeedyZero also achieves human-level performance, with a normalized mean of **1.0378** and a normalized median of 0.383, as shown in Table. 1. For more details about the experiment results, please refer to the paper.
>
> * **Question 5:** There are some issues with the language, and the authors should proofread their work again.
>
> Thanks for pointing out these language issues. We have updated our manuscript to correct them.

---

> > ### Comment · Reviewer_Rw6q · 2022-11-19
> > **Thank you!**
> >
> > The reviewers have addressed the majority of my concerns. I'm therefore raising my score to reflect this, and believe the codebase could be useful to the community towards rapidly developing on such MuZero-flavoured models, similar to how [1] has enabled rapid development on pixel-based continuous control.
> >
> > However I still have concerns around reproducibility. Concretely, it seems highly undesirable that the hardware itself affects the performance of the learned agent. This is compounded by the fact that hardware configurations will vary greatly between researchers, therefore it is possible to obtain results that people outside of certain research groups cannot replicate. This view is supported by the high variance across the 300k results, where as far as I can tell only hardware-level differences cause this. Indeed, unintended effects related to latency (e.g., server loads) may further act as hidden confounders for performance.
> >
> > [1] Mastering Visual Continuous Control: Improved Data-Augmented Reinforcement Learning, Yarats et al. ICLR22

---

### Official Review · Reviewer_Qh8r · 2022-10-25

**Confidence:** 4
**Correctness:** 3
**Technical Novelty And Significance:** 3
**Empirical Novelty And Significance:** 2
**Recommendation:** 6

**Clarity, Quality, Novelty And Reproducibility:**

Clarity:
- Medium-high. The paper is overall pretty well written.

Quality:
- Low. Comparing the proposed method and baselines with different sample complexity budgets is a big methodological flaw.

Novelty:
- Medium-high. The system-level optimizations are to my knowledge new. However, it's not clear how the absolute performance compares to existing methods.

Reproducibility:
- Low. The paper does not mention releasing code, the reproducibility checklist is absent, and all the code-level optimizations seem hard to reproduce from the paper alone.


**Strength And Weaknesses:**

Strengths
- The problem this paper is addressing is an important one: model-based methods, while known to be more sample efficient than model-free ones, are also known to be significantly slower to train. Therefore, speeding up these methods  would be useful both for the research community and for potential applications.
- The paper goes in depth into the system-level details of the algorithm, and identifies several inefficiencies, resulting in what seems like a very large speedup (10-20x) in wall-clock time

Weaknesses:
- A big problem with this work is that for some reason, they evaluate their algorithm with 300k steps from Atari, whereas previously published works use 100k (the standard). Therefore, despite the speed improvement, it's not at all clear how this algorithm does in terms of performance. I can't think of why this work would not use the standard 100k steps, unless the proposed algorithm would do worse than the others with 100k steps? This is very suspicious. I would suggest the authors use the standard 100k steps, and if they want to compare at 300k steps then they should rerun all the baselines at 300k as well.
- The proposed improvements are very much system-level, and I believe it will be hard for this work to be of use to others unless the code is released. However, there is no mention of code release, and the paper does not include a checklist which addresses questions of reproducibility.

**Summary Of The Paper:**

This paper proposes a distributed model-based RL algorithm building on MuZero/EfficientZero which is designed to reduce the wall-clock time required to train the agent. While EfficientZero is sample efficient, it still takes a long time to train. This work identifies and addresses several system-level inefficiencies, resulting in an algorithm called SpeedyZero which is evaluated on a modified version of the Atari 100k benchmark.







**Summary Of The Review:**

Overall, the problem is well-motivated and the proposed algorithm is interesting, but there are some major issues (inconsistent sample complexity budgets, unreleased code) which would need to be addressed for this paper to be acceptable for publication.


===========
In their new revision of the paper during the rebuttal period, the authors have added a comparison of their algorithm with 100k steps and uploaded their code. There is still a noticeable gap in performance between their algorithm and EfficientZero, but the speedup of their method is appreciated, and the fact that they release code could open the door to more community work in fast model-based RL methods. Therefore, I've updated my recommendation to weak accept.

---

> ### Author Response · Authors · 2022-11-15
> **Thanks for the review. We have updated the experiments to better support the claims in this paper and released the code for reproducibility.**
>
> We sincerely thank the reviewer for the valuable review and feedback. Here are our responses:
>
> * **Question 1:** A big problem with this work is that for some reason, they evaluate their algorithm with 300k steps from Atari, whereas previously published works use 100k (the standard). Therefore, despite the speed improvement, it's not at all clear how this algorithm does in terms of performance. I can't think of why this work would not use the standard 100k steps, unless the proposed algorithm would do worse than the others with 100k steps? This is very suspicious. I would suggest the authors use the standard 100k steps, and if they want to compare at 300k steps then they should rerun all the baselines at 300k as well.
>
> We apologize for the confusing experiment settings and presentation of results. We have updated the result of SpeedyZero under the 100k setting and updated the tables in the paper. When using 100k data, SpeedyZero also achieves human-level performance, reaching a mean score of **1.0378** and a median score of 0.383, as shown in Table. 1. For more details about the experiments, please refer to the paper.
>
> * **Question 2:** The proposed improvements are very much system-level, and I believe it will be hard for this work to be of use to others unless the code is released. However, there is no mention of code release, and the paper does not include a checklist which addresses questions of reproducibility.
>
> We have uploaded the full code of SpeedyZero in the zip file attached. Inside the zip file, the
> */SMOS-SpeedyZero* folder contains the shared memory object store used in SpeedyZero, and the */SpeedyZero* folder contains the implementation of SpeedyZero. For the three major system improvements mentioned in section 4.2, you can find the corresponding code in:
>
> 1. **Efficient On-Node Communication with Shared Memory:** This improvement is implemented in a standalone Python package, which can be found in the */SMOS-SpeedyZero* folder.
>
> 2. **Modular Design and Non-Trivial Workload Partition:** The code that implements our system partition can be found in */SMOS-SpeedyZero/core/train.py*. The code for internode communication is mainly in */SMOS-SpeedyZero/core/xnode.py*.
>
> 3. **Data Transfer Optimizations:** This optimization is more pervasive. Examples of data transfer optimizations can be found in */SpeedyZero/core/mcts.py* and */SpeedyZero/core/reanalyze_worker.py*.

---

> > ### Comment · Reviewer_Qh8r · 2022-11-17
> > **Thanks for the updates**
> >
> > Thank you for adding the requested experiments and uploading your code. I've raised by score from 3 to 6 as a result.

---

### Official Review · Reviewer_xxoB · 2022-11-04

**Confidence:** 2
**Correctness:** 3
**Technical Novelty And Significance:** 3
**Empirical Novelty And Significance:** 3
**Recommendation:** 6

**Clarity, Quality, Novelty And Reproducibility:**

The work seems original and novel. The writing is clear. I did not try to reproduce this work.

**Strength And Weaknesses:**

Strengths

1. The paper is well-written and easy to follow.
2. The proposed technique achieves significant speed up on the Atari benchmark.

Weaknesses

1. This work looks like a conglomeration of model-free and model-based RL methods. This is not necessarily a weakness, especially if this conglomeration was not obvious. I would like to understand this better.

**Summary Of The Paper:**

This paper proposes a a distributed RL system built upon a state-of-the-art model-based RL
method, EfficientZero, with system support for fast distributed computation. The paper also proposes a novel technique to stabilize massively parallel model-based training. Empirical evaluations demonstrate the efficacy of the approach.

**Summary Of The Review:**

I am not an expert in this area, and based on a quick search on this topic, I am not aware of any similar work. I am giving a rating of 6 for now. But, I can change my rating after reading the authors' rebuttal and discussing with the other reviewers.

---

> ### Author Response · Authors · 2022-11-15
> **Thanks for the review. We provide several related papers for your reference.**
>
> We sincerely thank the reviewer for the time in the review. We provide several related papers here for your reference.
>
> * **Question 1:** This work looks like a conglomeration of model-free and model-based RL methods. This is not necessarily a weakness, especially if this conglomeration was not obvious. I would like to understand this better.
>
> We clarify that our work is a model-based RL method since SpeedyZero is built on top of the model-based RL method EfficientZero and distributes the workflow in an efficient way to reduce wall-clock time. If the reviewer would like to know more about model-based RL methods, we recommend EfficientZero [1], MuZero [2], and SimPLe [3].
>
> References:
>
> [1] Ye W, Liu S, Kurutach T, et al. Mastering atari games with limited data[J]. Advances in Neural Information Processing Systems, 2021, 34: 25476-25488.
>
> [2] Schrittwieser J, Antonoglou I, Hubert T, et al. Mastering atari, go, chess and shogi by planning with a learned model[J]. Nature, 2020, 588(7839): 604-609.
>
> [3] Kaiser L, Babaeizadeh M, Milos P, et al. Model-based reinforcement learning for atari[J]. arXiv preprint arXiv:1903.00374, 2019.

---

> > ### Comment · Reviewer_xxoB · 2022-11-24
> > **My rating is kept**
> >
> > Based on the comments from other reviewers and the author responses, I decide to keep my rating as is: 6 (marginally above acceptance).

---

### Author Response · Authors · 2022-11-15
**We have updated the experiments in the paper and released the code to address the reviewers' concerns.**

We sincerely thank all reviewers' valuable reviews and feedback. We have updated the experiments and released the code. We have also revised the paper with the updates marked in red. The main changes include
* Experiments:
    * Results of SpeedyZero using 100k environment steps and 300k environment steps (16 seed)
    * The training time of EfficientZero on a machine with 8 x A100.
* Code Release:
    * SMOS, which is the shared memory object store we used in SpeedyZero.
    * SpeedyZero codebase.
* Paper:
    * Some language issues.
    * Revision of the experiment section with new results and clearer explanation.
    * More details about cluster hardware configuration in the appendix.

---

### Author Response · Authors · 2023-03-01
**Camera Ready Summary**

We sincerely thank all reviewers' and PC's efforts for reviewing our work and giving valuable feedbacks. We have updated the main paper and the appendix in the final camera ready version. The changes include:
- We introduce a new optimizer, Clipped LARS, to even further boost the performance of SpeedyZero with large batch size training.
- The main result is updated with large batch size training result. SpeedyZero is now able to achieve human-level performances on both the normalized mean and the normalized median with only **100k** environment steps and **50 minutes** of training time.
- Ablation study of Priority Refresh and Clipped LARS.
- Our early experiment results and discussions over "priority staleness" are moved into the appendix.

---

### Decision · Program_Chairs · 2023-01-20

**Decision:**

Accept: poster

**Justification For Why Not Higher Score:**

Uncertainty over robustness of results to changes in hardware or infrastructure

**Justification For Why Not Lower Score:**

Significant improvement in wall clock training time open-sourced enabling wider participation in model-based RL research

**Metareview: Summary, Strengths And Weaknesses:**

This paper presents SpeedyZero a distributed model-based RL training system that significantly improves wall-clock time when deployed on well matched hardware. After the rebuttal period, some questions remained regarding the variance of this improvement across different hardware and infrastructure choices. The open source release during the discussion period significantly raises the contribution, empowering the community to explore if these results hold across more commonly available compute resources.

**Note From Pc:**

if the above contains the word "oral" or "spotlight" please see: "oral" presentation means -> notable-top-5% and "spotlight" means -> notable-top-25%. As stated in our emails, we are disassociating presentation type from AC recommendations